# Hydrogen Isotopic Variations in the Shergottites

**Shuai Wang [1,2] and Sen Hu [1,3,\*]**

1 Key Laboratory of Earth and Planetary Physics, Institute of Geology and Geophysics, Chinese Academy of Sciences, Beijing 100029, China; wangshuai172@mails.ucas.edu.cn

2 University of Chinese Academy of Sciences, Beijing 100049, China

3 Innovation Academy for Earth Science, Chinese Academy of Sciences, Beijing 100029, China

\* Correspondence: husen@mail.iggcas.ac.cn; Tel.: +86-10-82998086

**Abstract:** Hydrogen isotopes in the shergottite Martian meteorites are among the most varied in Mars laboratory samples. By collating results of previous studies on major hydroxyl, deuterium, and $H_2O$ bearing phases, we provide a compendium of recent measurements in order to elucidate crustal-rock versus mantle-rock processes on Mars. We summarize recent works on volatile and $\delta D$ measurements in a range of shergottite phases: from melt inclusions, apatite, merrillite, maskelynite, impact melt glass, groundmass glass, and nominal anhydrous minerals. We interpret these observations using an evidence-based approach, considering two particular scenarios: (1) water-rock crustal interactions versus (2) magmatic-based processes. We consider the implications of these measurements and the scope they have for future studies, paying particular attention to future works on H, S, and Cl isotopes in situ, shedding light on the nature of volatiles in the hydrosphere and lithosphere of Mars.

**Keywords:** Mars; shergottites; water-rock interaction; hydrogen isotopes; water content

## 1. Introduction

Martian meteorites are the only samples of igneous basalts from Mars [1,2]. However, exceptions occur with the recent discovery of the regolith breccia, NWA 7034 [3,4]. Martian meteorites are not only crucial for constraining the petrology of Mars but also for studying the paleoclimate history [5], Martian chronology [6], the origin of Martian water, and the habitability of Mars [7,8]. Such records are however challenging to constrain based on the convolution of information retained in meteoritic phases from the limited suite of Mars laboratory samples we have on Earth.

Martian meteorites have been classified into regolith breccia (e.g., NWA 7034 [3]), orthopyroxenite (ALH 84001), clinopyroxenites (nakhlites), dunites (chassignites), and the most abundant Martian basalt—the shergottites [2]. The shergottites are subclassified into basaltic (e.g., Shergotty), olivine-phyric (e.g., Tissint), and poikilitic (e.g., ALH 77005) sub-groups based on texture and mineralogy [9–20]. They are further divided into depleted, intermediate, and enriched—three geochemical groups based on the relative rare earth elemental (REE) abundances and their isotopic compositions of Sr, Nd, and Hf [1,12,21–27]. The depleted shergottites crystallized under reducing conditions (Log $fO_2$ QFM ~ −3) that depleted their light rare earth elemental (LREE) content [28–30]. They also have low initial $^{87}Sr/^{86}Sr$ isotopic ratios [28,29], and high initial $^{143}Nd/^{144}Nd$ and $^{176}Hf/^{177}Hf$ isotopic ratios [21]. The enriched shergottites crystallized under more oxidizing conditions (Log $fO_2$ QFM ~ −1 to ~0) and have flat REE patterns [28–30], displaying geochemical characteristics opposite to their depleted counterparts. The intermediate shergottites are those with geochemical and redox conditions intermediate between the enriched and depleted shergottites. Two models have been proposed to interpret the variation in the REE patterns, Sr, Nd, and Hf isotopic compositions, and redox conditions: (1) distinct mantle sources and (2) variable to enriched, oxidized crustal material assimilation of an original depleted parental magma [1,13,25,30–34].



The crystallization ages of Martian meteorites vary from: (1) ~4.4 Ga for various igneous clasts from the regolith breccia NWA 7034 and its pairings [4,35,36] based on U-Pb dating of zircons, (2) ~4.1 Ga for the orthopyroxenite ALH 84001 [37], (3) ~1.3 Ga for clinopyroxene nakhlites, and dunitic chassignites [6], and (4) ~600-150 Ma for most shergottites based on bulk Rb-Sr and Sm-Nd analyses [6], Pb-Pb isochrons measured in situ [38], and U-Pb dating of baddeleyite and phosphates [39–42], except for the two recent identified shergottites NWA 8159 and NWA 7635 which are dated at ~2.4 Ga [19,20]. Around 80% of Martian meteorites are shergottites (Meteoritical Bulletin). Bouvier et al. [43,44] on the other hand proposed that the depleted shergottites are ~4.3 Ga old, whilst the enriched shergottites are as old as the orthopyroxenite ALH 84001 (~4.1 Ga [37]) based on the Pb-Pb isochron. However, most recent in situ Pb-Pb analyses are in good agreement with previous Rb-Sr, Lu-Hf, and Sm-Nd ages, confirming the young ages of the enriched shergottites [38]. Although this review does not rigorously discuss the chronology of the SNC (shergottites, nakhlites and chassignites) Martian meteorites, the epoch of Mars relating to the shergottite parent body formation and evolution will consider their young crystallization ages in this paper.

The red planet is thought to have been a warm and wet world in its first hundred million years [45]. This is supported by, for instance, abundant phyllosilicates in ancient terrains [46], fluvial outflow channels [45], and sedimentary outcrops [47]. However, over geological timescales, the red planet transformed into an arid and cold world since the late Hesperian (~3.2 Ga) [48]. This can be explained by the loss of Martian water to space via solar wind bombardment during the decay of its magnetic field [48], enriching its atmosphere in $\delta$D (4950 ± 1080‰ [49]) as observed in situ and by FTIR measurements of the Martian atmosphere—significantly higher than the initial $\delta$D of ~0‰ [50–54]. The current hydrogen isotopic composition of the Martian atmosphere is thought to have equilibrated with surficial/underground water via exchange between the Martian hydrosphere and lithosphere [55–57].

A meteorite should equilibrate with its parent body in hydrogen isotopic compositions during crystallization (~0‰, [50–53]). However, most of the shergottites display $\delta$D values higher than that presumed upon basaltic crystallization (up to ~4000–6830‰) [35,51–53,58–75]. Such high values are inconsistent with parent magma degassing on Mars [76]. Crustal assimilation during magma ascent and eruption [58], post-emplacement water-rock interaction [67,77], and impact-induced water-rock interactions [60,65,68] provide plausible explanations for these H isotopic compositions. The recent hydrogen isotopic datasets on Martian meteorites [68], as well as other isotopic systematics of e.g., O [78], Cl [79–81], and S [82], tend to support that isotopic exchanges occurred between Martian minerals and the surficial water/fluid on the near surface of Mars.

The Martian igneous basalts often contain alteration mineral assemblages. Bridges et al. [83], for instance, reviewed the secondary mineral assemblages in the SNCs and discussed the implications of near-surface alteration processes on Mars. Secondary phases, such as iddingsite, illite, gypsum, and smectite are found in the nakhlites [83]. In contrast, a few hydrous alteration minerals (salts, calcite, smectite, etc.) have been observed in the shergottites, although they are probably terrestrial in origin [83–88] (see Section 3). Although the shergottites lack any unambiguous signs of Martian mineral alteration, their highest $\delta$D values among the SNCs in both bulk (e.g., ~2100‰ for Shergotty vs ~800‰ for ALH 84001) [89] and in situ (up to ~4000–6830‰ for shergottites *vs* <3000‰ for non-shergottites) [35,51–53,58–75], is consistent with the progressive loss of Martian water over time [5,90].

In this paper, we mainly focus on the progresses made on the hydroxyl, water, and hydrogen isotopic content of different phases in the shergottites, namely: apatite, merrillite, melt inclusions, maskelynite, impact melt glass, groundmass glass, and nominal anhydrous minerals. We aim to collate various measurements and studies in order to explore water-rock verses magmatic H isotopic processes.

## 2. Water Reservoirs on Mars

Today, the red planet currently has an active seasonal cycle of ice caps, permafrost, and sub-surface terrains through seasonal cycles. In the shergottites, measurements of the water content and H isotopes

in apatite, amphibole, melt inclusion glasses, impact melt glasses, and anhydrous silicates display two distinct water reservoirs on Mars [50–53,58,60,67–72,91,92]: The Martian mantle has a low $\delta D$ value of ~0‰, whilst the Martian surficial/crustal water reservoir is higher at ~5000–6000‰, which probably isotopically equilibrated with the Martian atmosphere ($\delta D$ = 4950 ± 1080‰) [49,56,57]. Usui et al. [72] proposed an intermediate water reservoir ($\delta D$ ~ 3000‰) according to the relationship between $H_2O$ and $\delta D$ of groundmass glasses from Y 980459. However, it is still not clear to what degree of mixing occurred between the mantle and surficial water [50,68]. Barnes et al. [93] proposed that multiple water reservoirs formed early in the interior of Mars to interpret the variation in the hydrogen isotopic composition of different Martian meteorites. The D/H ratios of the Martian crust, enriched shergottites mantle source, and depleted shergottites mantle source are estimated to be 2.68–5.73 × $10^{-4}$, 8.03 ± 0.52 × $10^{-4}$, and 1.99 ± 0.02 × $10^{-4}$ respectively [93]. However, this model cannot account for the large differences in the H isotopic composition of the mantle source where H should be highly equilibrated under high pressure and temperature conditions because of the fast diffusion rate of H [94].

Unlike that of the H isotopic composition of Martian water, the cause of the $H_2O$ content in H isotopic bearing minerals is arguably less constrained [95,96]. This is because $H_2O$ is easily affected by magma degassing [97], shock metamorphism [98], post-crystallization hydrothermal activity [67], and terrestrial contamination [99]. McCubbin et al. [96] summarized the distribution of $H_2O$ in the Martian mantle, calculating the volatile compositions of melt inclusion glasses, apatite, and amphiboles from Y 980459 [53], QUE 94201 [95], Shergotty [95], and Chassigny [100]. They estimated that the enriched shergottite source has 36–73 ppm $H_2O$, and the depleted source has 14–23 ppm $H_2O$ [96]. Filiberto et al. [50,95] reviewed the volatiles in the Martian interior based on bulk chemistry of Martian meteorites and the mineral chemistry of Martian apatite and amphibole. They estimated that the Martian mantle has Cl content similar to the terrestrial enriched mantle, and F and water content similar to the terrestrial primitive mantle [50,95]. The C content of the Martian interior is currently not well-constrained compared with other volatile elements [50,101]. However, the contribution of surficial volatiles, especially D- and $^{37}$Cl-rich water/fluid, has not been carefully considered to constrain the volatile abundances for the Martian mantle. For instance, the impact melt glasses of Tissint are D-rich (up to 4867‰), implying a contribution from the Martian surficial water reservoir [60]. The positive correlation between the water and Cl content measured in impact melt glasses from Tissint [60] and melt inclusion glasses from NWA 6162 [68] also indicate that these Martian rocks retain some Cl sourced from the Martian subsurface water reservoir. It is also suggestable that the Martian crustal reservoir had Cl contributions from various sources [35,79–81,102]. Additional studies estimated the concentration of water in the Martian crust to vary from 350 ppm to 1.3 wt. % based on gamma ray spectroscopy and Martian meteorites studies [3,103]. McCubbin et al. [96] estimated that bulk silicate in the Martian crust contains ~137 ppm water. Recent studies have shown that Mars has subglacial liquid water in the North Polar Regions [104]. It is believed that the Martian crust, especially the upper surface of Mars, is water- and volatile-rich in e.g., F, Cl, S, and C [50,95,96]. The Martian atmosphere displays geographical variation in water content from ppb levels in the winter up to 250 ppm in the late spring, based on remote telescopic observations [90], implying seasonal variation of volatile elemental isotopes.

## 3. Alteration Minerals in the Shergottites

Bridges et al. [83] reviewed the secondary mineral assemblages in the SNCs. Here we briefly summarize observations made on alteration minerals in the shergottites in the recent past. Taylor et al. [85] reported some calcite, gypsum, celestite, Fe hydroxides, and smectite in Dhofar 019. They argued that some smectite cut and intersected by micro-faulting of maskelynite were pre-terrestrial [85]. Hallis et al. [84] on the other hand studied alteration phases in Dhofar 019 by TEM and interpreted them as terrestrial based on the presence of celestine within the orangette layers, the absence of shock dislocation features within calcite, and the Mg-rich nature of the phyllosilicate. Gnos et al. [86] found secondary Mg-Fe-Si silicate and gypsum crystals in SaU 094. Some salt



assemblages—perchlorate, chlorate, and nitrate—consistent with Martian surface/near surface fluidic processes, were identified in Shergotty and EETA 79001 [105–107]. Some jarosite veins were reported in QUE 94021 [108]. Their δD values vary from −386‰ to −325‰, which is significantly lower than both the Martian mantle (~0‰) and Martian crustal water reservoirs (~5000–6000‰) [49,51,56,67]. This suggests the formation of jarosite in Antarctica. Changela et al. [109] also identified jarosite cross-cutting the fusion crust of nakhltie Y000593, suggesting a terrestrial origin. Alternatively, they may have formed on Mars and then re-equilibrated isotopically with terrestrial water. Kuebler [87] reported the jarosite and iddingsite alteration in ALH 77005, suggesting a hydrothermal event on Mars. Piercy et al. [88] conversely observed olivine grains from an olivine-phyric Martian shergottite NWA 10416, showing orange-brown altered cores (Fo-rich) and clear, unaltered rims (Fa-rich). The more susceptible Fo-olivine was interpreted to have altered in Northwest Africa [88] and a cross-cutting shock melt vein was altered, suggesting alteration of the meteorite post-ejection from Mars. In summary, the shergottites have ambiguous interpretations for the origin of alteration minerals compared unlike e.g., the nakhlites [3,83,110]. This implies that the shergottites underwent minimal hydration when compared with the other Martian meteorites [83,110]. This is consistent with the paleo-climatic age of the shergottite lithology on Mars; the younger shergottite Martian environment was already minimal in surficial water [6,48].

## 4. Evidence of Subsurface Water-Rock Interactions on Mars in the Shergottites

Almost all phases in the shergottites, e.g., apatite, merrillite, melt inclusions, maskelynite, and impact melt glasses, have δD values up to ~4000–6830‰ [35,51–53,58–73]. A summary of water content and hydrogen isotopic compositions of the shergottites is listed in Table 1. We focus on discussing the results from melt inclusions and apatite that are the major H bearing phases in the shergottites.

**Table 1.** A summary of water content and H isotopes analysis of shergottites.

| Meteorite | Groups [1] | Mineral/Phases | $H_2O$ (ppm) | δD (‰) | Ref. |
|---|---|---|---|---|---|
| Shergotty | BS | Apatite | 3000–7000 | 2953–4606 | [51] |
| | | Kaersutite | 1000–2000 | 512 | [58] |
| | | Augite | 323 | | [51] |
| | | Olivine | 12–86 | | [51] |
| | | Pigeonite | 800–1360 | −153–−60 | [59] |
| | | Silica | 179–267 | 1246–1975 | [59] |
| | | Mesostasis | 111–662 | 441–490 | [59] |
| | | Post-stishovite | 30–39 | 1246–1975 | [111] |
| Zagami | BS | Kaersutite | 1000–2000 | 1498–1672 | [58] |
| | | Apatite | 3000–4000 | 2962–4358 | [58] |
| | | Mesostasis | 338–624 | 285–1195 | [59] |
| | | Maskelynite | 12–71 | 579–2532 | [59] |
| | | Silica | 30–183 | 1173–2704 | [59] |
| | | Pigeonite | 3360 | −204 | [59] |
| QUE 94201 | BS | Apatite | 2200–6400 | 1683–3565 | [62,73] |
| Los Angeles | BS | Apatite | 1800–6200 | 2794–4348 | [61,62] |
| EETA 79001B | BS | Apatite | 1160 | 146 | [59] |
| | | Maskelynite | 111–128 | 1540–1589 | [59] |
| | | Olivine | 29 | 1303 | [59] |
| | | Pyroxene | 92 | −26 | [59] |
| EETA 79001C | IMG | Mafic glass | 101–556 | 2289–2901 | [59,71] |
| | | Maskelynite | 4–98 | −91–3938 | [59,71] |
| | | Impact melt glass | 90–646 | 3368–4639 | [71] |
| | | Olivine | 4–7 | −146–−55 | [71] |
| | | Pyroxene | 10–41 | 1729–2837 | [71] |
| EETA 79001A | OS | Impact melt glass | 232–393 | 1454–1644 | [72,112] |

**Table 1.** *Cont.*

| Meteorite | Groups [1] | Mineral/Phases | $H_2O$ (ppm) | δD (‰) | Ref. |
|---|---|---|---|---|---|
| Tissint | OS | Melt inclusions | 1500–5629 | −98–1397 | [52,65] |
| | | Maskelynite | 30–6200 | −222–3682 | [52,60] |
| | | Olivine | 50–1273 | −149–470 | [52,65] |
| | | Clinopyroxene | 1288 | −131 | [65] |
| | | Merrillite | 50–9600 | −105–2418 | [52] |
| | | Impact melt glass | 179–2388 | 45–4867 | [60,70] |
| | | Ringwoodite | 714–1132 | 3834–4224 | [65] |
| DaG 476 | OS | Maskelynite | 40–1110 | 352–2347 | [59] |
| SaU 005 | OS | Maskelynite | 32–76200 | −105–3260 | [59] |
| Y 980459 | OS | Melt inclusions | 146–841 | −95–285 | [53,72] |
| | | Groundmass glass | 17.7–257.4 | −71–1562 | [53,72] |
| LAR 06319 | OS | Melt inclusions | 65–1872 | 1150–6830 | [53,69,72] |
| | | Impact melt glass | 117.8–163.8 | 2096–2929 | [72] |
| | | Apatite | 2854–9964 | 3340–4380 | [69] |
| | | Merrillite | 204–812 | 1070–5260 | [69] |
| NWA 6162 | OS | Melt inclusions | 11–2421 | −560–6137 | [68] |
| | | Maskelynite | 18–181 | −426–4601 | [68] |
| | | Fusion crust | 13–37 | −728–1889 | [68] |
| ALH 77005 | PS | Melt inclusions | 0.74–1770 | −106–304 | [59] |
| | | Mafic glass | 105–520 | 1301–3030 | [59] |
| | | Maskelynite | 61–930 | 982–4214 | [59] |
| | | Merrillite | 6740 | 22 | [59] |
| | | Pigeonite | 360 | 69 | [59] |
| | | Olivine | 0.53 | 35 | [59] |
| GRV 020090 | PS | Melt inclusions | 91–10308 | 3386–6034 | [67] |
| | | Apatite | 1020–5762 | 737–4239 | [67] |
| Y 980428 | PS | Pyroxene | 12–15 | 403–522 | [113] |
| | | Olivine | 23 | 262 | [113] |
| GRV 99027 | PS | Apatite | 1200–4300 | 1326–4064 | [63] |
| | | Merrillite | 500–1100 | 2153–4745 | [63] |

[1] BS: basaltic shergottite, OS: olivine-phyric shergottite, PS: poikilitic shergottite, IMG: impact melt glasses.

## 4.1. Melt Inclusions

Melt inclusions in igneous minerals can provide constraints on the magma compositions, especially for planetary samples where mass is severely limited [53,67,68,72,114–117]. Most melt inclusions from the shergottites probably experienced variable degrees of post-entrapment crystallization. This is supported by the observation of layers of Ca–Al-rich pyroxene overgrowths along the outer rims of primary igneous minerals as well as some other small euhedral pyroxene crystals enclosed by silica [52,67,68,114]. Moreover, parent magma degassing [118], ascent/eruption degassing [76,97], post-emplacement water-rock interaction [67,77], secondary aqueous alteration [66], shock metamorphism [60,65,98], and terrestrial alteration [99] could also affect the water content and H isotopic compositions in the melt inclusions in non-mutually exclusive ways. Thus explanations for the H isotopic content in melt inclusions are complicated by a range of geochemical processes.

Poikilitic Shergottites: Boctor et al. [59] reported the water content and H isotopic compositions of melt inclusions from ALH 77005 at 0.74 ± 0.13 to 1770 ± 200 ppm and −106 ± 15 to 304 ± 67‰ respectively. Hu et al. [67] also measured the H isotopes (3386 ± 126 to 6034 ± 72‰) and water content (90 ± 5 to 10300 ± 1222 ppm) of melt inclusions from GRV 020090. The $H_2O$ content positively correlates with δD along a two end-member mixing trend (Figure 1). The D-enriched end-member with a δD value of ~5000–6000‰ could represent the Martian surficial/crustal water reservoir. The D- and $H_2O$-poor end-member could be the Martian mantle composition. However, this requires the consideration of degassing processes during crystallization [67,68] which are poorly constrained. Melt inclusions in olivine from GRV 020090 show diffusional hydration profiles, with both the water content and the δD values decreasing from 5337 to 170 ppm and from 5519 to 3386‰ at the rims towards the cores respectively, suggesting that Martian D in surficial water entered into the melt inclusions via

diffusion [67]. The duration of subsurface liquid water on Mars can reach up to 130,000–250,000 years based on diffusion simulations [67].

Olivine-phyric Shergottites: Y 980459 melt inclusions measured by Usui et al. [53], the most primitive shergottite, has a water content varying from 146 to 841 ppm (assuming $H_2O$ (ppm) = H/Si) with similar $\delta D$ of ~270‰. The melt inclusions from Tissint range from 1500 to 5629 ppm and $\delta D$ from −98 to 1397‰ [52,65]. In contrast to Y 980459 and Tissint, a melt inclusion from LAR 06319 [53] measures at 1872 ppm with a $\delta D$ of 5079‰, also plotting in the two end-member mixing trend of GRV 020090 (Figure 1). Koike et al. [69] reported the water content in melt inclusions from LAR 06319 to have 65–821 ppm and a $\delta D$ of 1150 ± 670 to 6830 ± 460‰. The melt inclusions from NWA 6162, another depleted olivine-phyric shergottite, probably paired with SaU 005, measures ~0–2421 ppm water and a $\delta D$ of ~0–6137‰ [68], also plotting along the similar two end-member mixing trend as GRV 020090 (Figure 1).

Melt inclusions from poikilitic and olivine-phyric shergottites display two distinctive petrogenesis (Figure 1). Most of the analytical spots plot along the two end-member mixing trend and the other spots seem to be affected by a D-poor end-member ($\delta D$ ~ 0‰) either from degassing water on Mars [35,51] or terrestrial contamination [99] (Figure 1). The diffusional profiles of D-enriched water into the melt inclusions from GRV 020090 [67] and dendritic shapes of $H_2O$– and D-enriched features within the melt inclusions from NWA 6162 [68] suggest that Martian D in water from surficial/crustal water reservoirs interacted with the parent rock postdating crystallization on Mars. This may have been heated by volcanic intrusions or impact [67,68]. The melt inclusions from ALH 77005, Y 980459, and Tissint [53,59,65] plot along a reverse trend between $H_2O$ and $\delta D$ values, contrary to that of GRV 020090, NWA 6162, and LAR 06319 [67–69] (Figure 1). This may be the result of either an interaction of Martian magmatic water on Mars [35,51] or terrestrial water on Earth [52,99]. As the Martian mantle and terrestrial water are thought to have similar $\delta D$ values (~0‰) [51–53], determining the source of the D-poor end-member solely based on $H_2O$ and $\delta D$ is ambiguous. More work is required for better constraining this issue, e.g., Cl [79–81,102], S [82], and O [78] isotopes, as well as other volatile elements (C, F, S, Cl) [50,68,95,102].

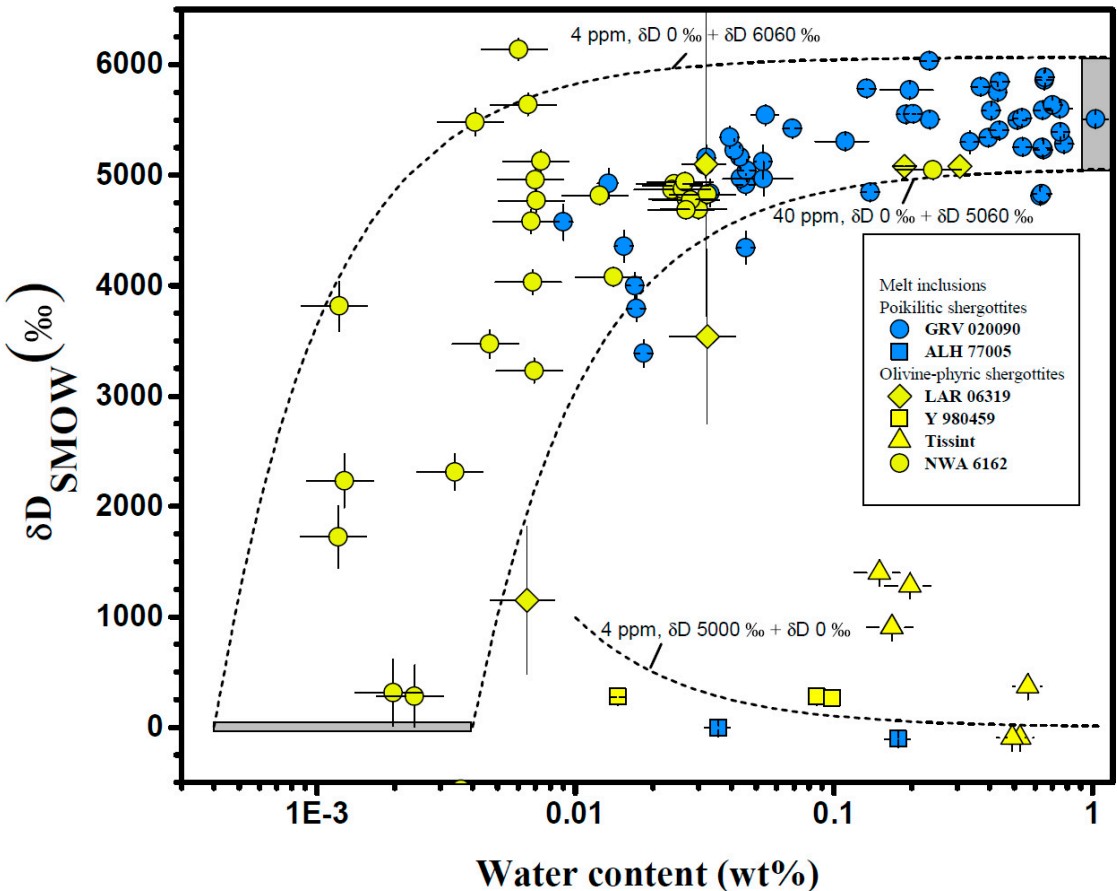

**Figure 1.** The relationship between water content and δD values of melt inclusions from different shergottites. The dashed lines represent the mixing of the post-erupted water ($H_2O$ = 4–40 ppm and δD ~ 0‰, lower left hatched area) and the Martian crustal water (δD = 6060–5060‰, upper right hatched area) from [68]. GRV 020090 from [67], ALH 77005 from [59], LAR 06319 from [53,69,72], Y 980459 from [53,72], Tissint from [52], and NWA 6162 from [68].

### 4.2. Apatite

Apatite is a major OH–, F–, and Cl–bearing accessory mineral in Martian meteorites. It has been used for estimating the volatile concentration of their parent magmas using partition coefficients [32,100,119]. However, recent synthetic experimental studies indicate that the OH–, F–, and Cl–content of apatite are unpredictably dependent on the nature of the melt and are not reliable for estimating the volatile abundances of the parent magma [120,121]. As apatite is a late crystallization phase, the melt from which apatite formed is likely to incorporate some Martian upper crustal materials [122]. Therefore, apatite is probably not a suitable proxy for constraining the volatile abundances of the Martian mantle. However, apatite is a crystalline phase unlike the quenched glasses in melt inclusions and was thus altered less in any subsequent hydrothermal processes.

Poikilitic Shergottites: Three SIMS analytical spots of apatite from GRV 99027, a depleted shergottite, have from 0.12 to 0.43 wt. % water and δD values from 1300 ± 193 to 4064 ± 283‰, in which two spots probably were partially covering merrillite co-occurring with apatite [63]. GRV 020090's water content ranges from 1020 ± 248 to 5762 ± 755 ppm and δD values from 737 ± 107 to 4239 ± 81‰ in apatite [67]. Analyses of apatite from GRV 020090 and GRV 99027 plot along a similar positive $H_2O$ verses δD trend (Figure 2), which can be interpreted by progressive fractional crystallization, causing a gradual increase in water content, and the assimilation of D-enriched crustal materials [67,122].

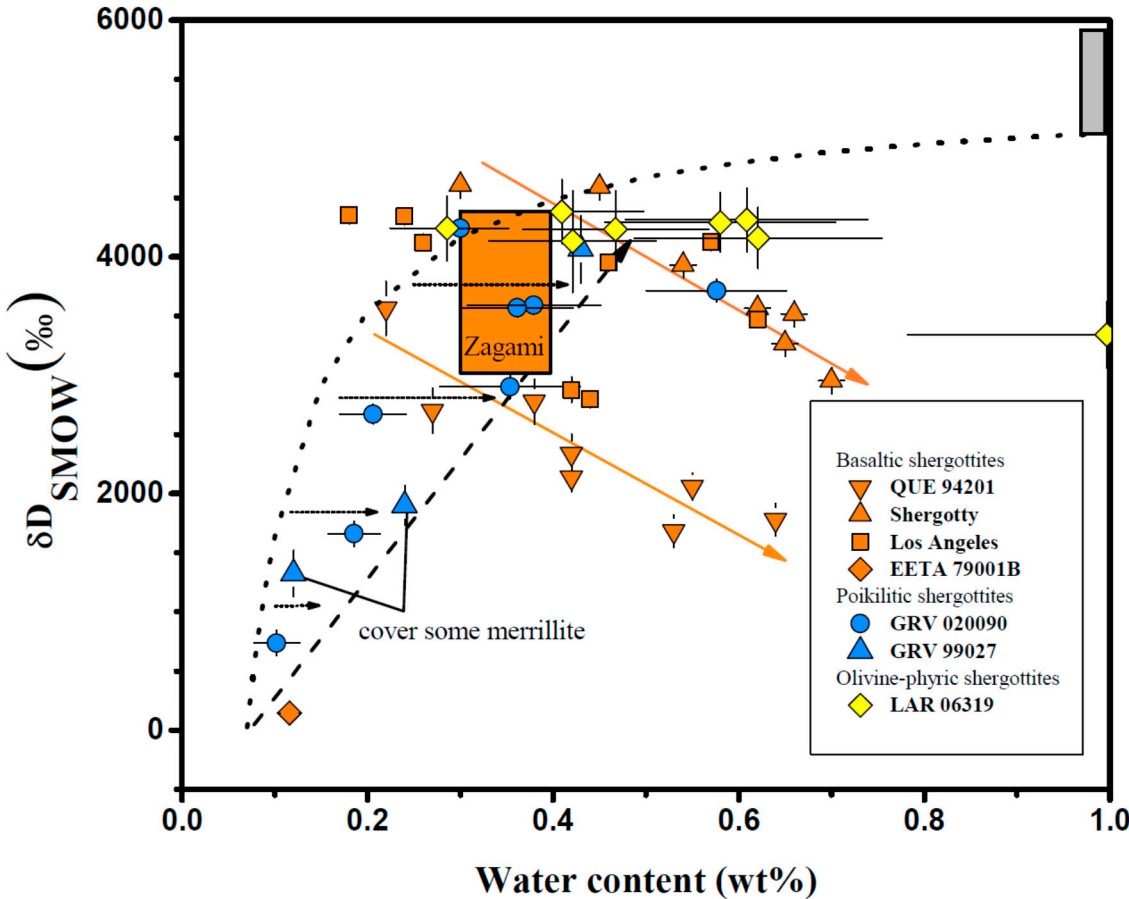

**Figure 2.** The relationship between water content and δD values in shergottite apatite grains. Poikilitic shergottites, GRV 020090, and GRV 99027, display a positive trend. In contrast, basaltic shergottites, QUE 94201, Shergotty, Los Angeles, and EETA 79001B show a negative correlation between $H_2O$ and δD. Most apatite grains from olivine-phyric shergottite LAR 06319 have similar δD values with water content varying from ~0.3 to ~0.6 wt. % except for a spot having a water content of ~1.0 wt. % and a δD of ~3300‰. The dash line shows the combined effect of the fractional crystallization (horizontal arrows) and assimilation (dot line) of Martian crustal materials with magma ascension [67]. QUE 94021 from [73], Shergotty from [51,61], Los Angeles from [61], EETA 79001B from [59], probably covering on adjacent mafic phases, GRV 020090 from [67], GRV 99027 from [63], LAR 06319 from [69], Zagami from [58].

Basaltic Shergottites: Four basaltic shergottites, QUE 94201 [73], Shergotty [51,61], Zagami [58], and Los Angeles [61], display two negative trends opposite to the positive trends of the poikilitic shergottites (Figure 2). One spot from EETA 79001B with both low $H_2O$ (1160 ± 100 ppm) and δD (146 ± 25‰) probably overlapped surrounding silicates [59]. $H_2O$ content in apatite from QUE 940201 varies from 2200 to 6400 ppm, negatively correlating with δD values (1778 ± 136 to 3565 ± 228‰) (Figure 2) [73]. Shergotty ($H_2O$: 3000–7000 ppm, δD: 2953–4606‰) [51,61], Zagami ($H_2O$: ~3000–4000 ppm, δD: 2963–4358‰), and Los Angeles ($H_2O$: 1800–6200 ppm, δD: 2794–4348‰) [61] have similar negative trends between $H_2O$ and δD with QUE 94201 but with relatively higher δD values, with exception for two spots from Los Angeles (Figure 2). The datasets of QUE 94021 suggest a mixture of two end-members, and plausibly indicate the addition (or exchange) of atmospheric D/H water (δD ~4200‰ [54]) with initial water-bearing minerals with a δD of ~900 ± 250‰. This suggests that the D/H value of Martian magmatic water is ~twice the value of terrestrial water [73]. This scenario requires multiple episodes on Mars: firstly an increase in δD of apatite from ~0‰ [51–53] in the Martian mantle to ~4000‰ from the Martian crustal/surficial water reservoir [54]. Crystallization from a D-rich parent magma [67] or post-crystallization hydrothermal event [77], followed by another

hydrothermal episode may have lowered the $\delta$D value to ~900‰ and increased the water content in apatite [35]. Alternatively, terrestrial contaminations and/or contributions from cracks in apatite can also account for the negative correlation between $H_2O$ and $\delta$D in apatite grains from QUE 94021, Shergotty, Zagami, and Los Angeles (Figure 2) [52,65,99].

Olivine-phyric Shergottites: Most apatite grains from olivine-phyric shergottite LAR 06319 have similar $\delta$D values (average: 4249 ± 89‰), distinct from the negative trends defined by QUE 94021, Shergotty, Zagami, and Los Angeles, with water content varying from ~0.3 to ~0.6 wt. %, except for a spot having at~1.0 wt. % water and a $\delta$D value of ~3300‰ [69] (Figure 2). The relatively homogeneous and high $\delta$D values of most apatite grains in LAR 06319 suggests that they isotopically equilibrated with the Martian crustal/surficial water, with a partial form of hydration postdating the crystallization [69]. This is consistent with the post-crystallization interaction between Cl– and $H_2O$–rich crustal fluids that may have interacted in LAR 06319 apatite by Howarth et al. [77].

### 4.3. Merrillite

Merrillite ($Ca_{18}Na_2Mg_2(PO_4)_{14}$) is a common water- and halogen-free accessory phosphate in Martian meteorites [123]. It typically co-occurs with apatite in most poikilitic and basaltic shergottites and is widespread among Martian meteorites [2]. McCubbin et al. [123] reported some whitlockite ($Ca_9(Mg,Fe^{2+})(PO_4)_6[PO_3(OH)]$) components in merrillite from Shergotty. Therefore, merrillite is a useful analytical target for identifying whitlockite and monitoring the secondary processes on Mars.

Overall, the water content of merrillite in all types of shergottites negatively correlates with $\delta$D regardless of their petrogenesis, along the two end-member mixing trend (Figure 3). The water content (~0.3–0.96 wt. %) of spots of merrillite from Tissint with low $\delta$D values (−105–524‰) was measured from a polished thin [52]. In contrast, the anhydrously prepared thick section of Tissint has significantly lower water content (mostly lower than 0.2 wt. %) and higher $\delta$D values (272–2418‰) [52]. These distinctive features are a suggestible analytical artifact during sample preparation [52]. In comparison, merrillite from GRV 99027 [63], LAR 06319 [69], and EETA 79001C [71] have water content lower than ~0.1 wt. % and large $\delta$D variation from ~1500 to 5200‰ (Figure 3). Moreover, the detailed analyses of merrillite from EETA 79001C, an impact melt lithology, shows that the D-enrichment in merrillite probably relates to the distance to the melt pocket, suggesting that the D-rich signature of merrillite formed during the shock event [71] rather than by crystallization from the parent melt [58,61]. In contrast to the significant reverse trend between $H_2O$ and $\delta$D in Tissint and ALH 77005 merrillite [52,59], merrillite from GRV 99027 [63], LAR 06319 [69], and EETA 79001C [71] does not measure terrestrial water or degassing water from Martian mantle (Figure 3). This indicates that most merrillites in Martian meteorite should have a D-rich nature except for those spots from Tissint showing strong contamination during sample preparation reported by Mane et al. [52] and ALH 77005 [59]. McCubbin et al. [123] reported that some whitlockite components in merrillite from Shergotty derived via dehydration processes. Dehydration may play a role in outgassing H and enriching D. However, degassing of water usually results in fractionation of D less than 500‰ based on thermodynamic simulation [76] and synthetic experiments [97]. Alternatively, such distributions can be explained by a mixing of Martian mantle signature ($\delta$D ~ 0‰) with D-enriched Martian crustal water reservoir when ignoring those spots with significantly high water content and low $\delta$D values (Figure 3).

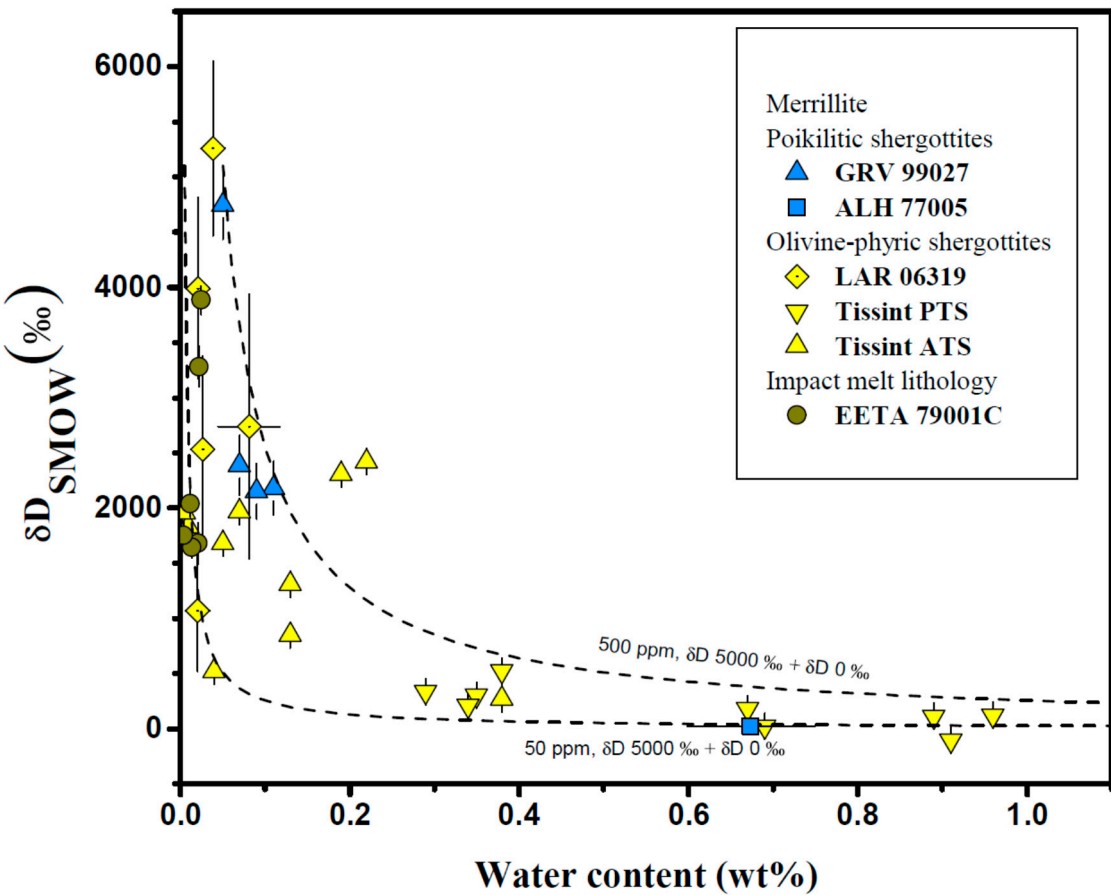

**Figure 3.** The relationship between water content and δD values of the merrillite grains from Martian shergottites. All of the datasets depict a negative trend regardless of petrographic type. GRV 99027 from [63], ALH 77005 from [59], LAR 06319 from [69], Tissint from [52], and EETA 79001C from [71]. PTS: polished thin section, ATS: anhydrously prepared thick section.

### 4.4. Maskelynite

Maskelynite or feldspathic glasses, the quenched glasses of plagioclase by shock, is a very common, widespread and texturally smooth phase in the shergottites. The water content of maskelynite in the literature is reported to vary from ~1 ppm to 7.6 wt. % [52,59,60,68,70,71,124,125] with δD varying from ~−400 to ~4500‰ (Figure 4). However, most analytical spots on maskelynite have water content lower than 200 ppm, with a large variation in δD values (inset diagram in Figure 4). This is consistent with the bimodal mixing of Martian mantle and surficial water reservoirs defined by the melt inclusions in Figure 1 [67,68]. Several spots having high water content with relatively low δD values were probably attributed to some degree of terrestrial contamination or analyses along cracks, especially for the spots with water content higher than 6000 ppm [59]. Moreover, two profiles measured on the smooth maskelynite from NWA 6162 show that both $H_2O$ and δD increase from the cores towards the rims, suggesting that D-enriched water from Martian crustal/surficial water reservoir diffused into maskelynite after/during the plagioclase transformation into maskelynite by impact [68].

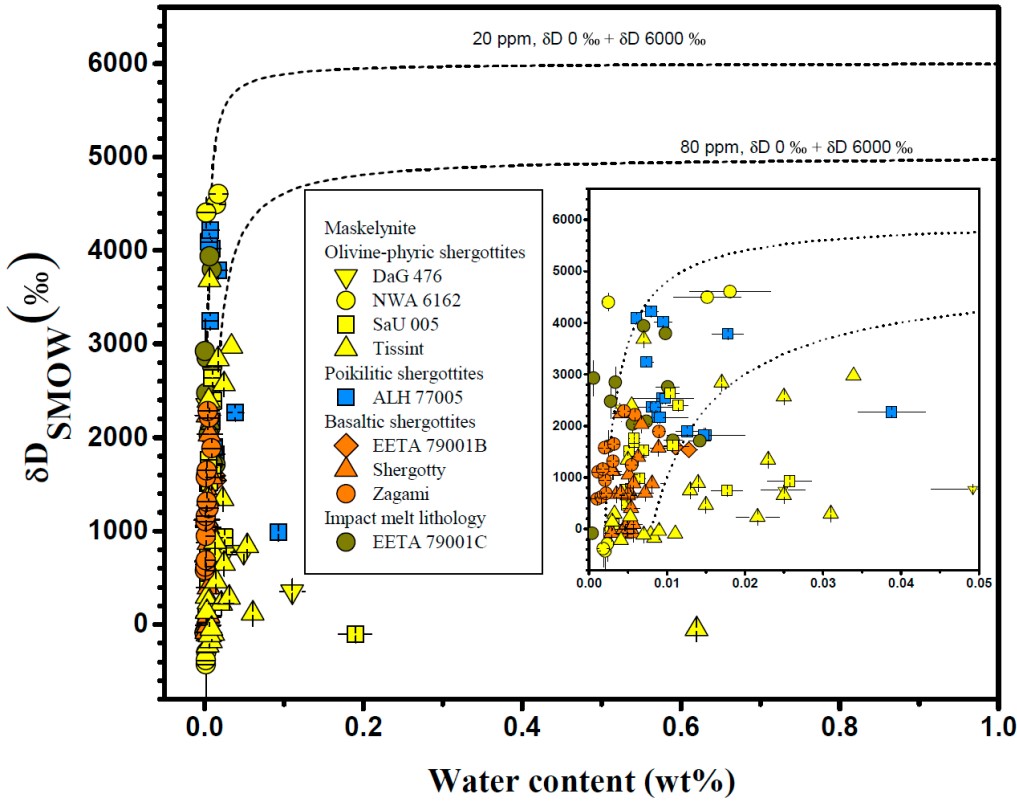

**Figure 4.** The relationship between water content and δD values of shergottite maskelynite. Most shergottite maskelynite has water content less than 200 ppm but significantly large variation in δD values, plotting along a two end-member mixing trend, except for several grains displaying a negative correlation between the water content and δD values. The inset diagram magnifies the data ranging from 0 to 500 ppm. The dashed lines show the mixing of the post-erupted water ($H_2O$ = 20–80 ppm and δD ~ 0‰) and the Martian crustal water (δD = 6060–5060‰) from [68]. Tissint from [52,60], EETA 79001C from [71], NWA 6162 from [35], Shergotty and Zagami from [125], ALH 77005 from [124], and the other literatures are from [59].

### 4.5. Impact Melt Glasses and Groundmass Glasses

Impact melt glasses, common in shergottites, are quenched melts of the host rock induced by shock metamorphism on Mars [126]. In contrast, groundmass glasses, such as those mostly found in Y 980459, are quenched residual parent melts because of rapid cooling rates [127]. Impact melt glasses and groundmass glasses have distinctive origins but share a common amorphous structure. They both record water-rock signatures and are helpful for constraining the nature of post-crystallization hydration on Mars.

The water content and δD values of impact melt glasses from olivine-phyric shergottites Tissint, EETA 79001A, and LAR 06319, poikilitic shergottite ALH 77005, and impact melt lithology EETA 79001C have been measured in situ [59,60,65,70–72] and are plotted in Figure 5. Three spots from Tissint reported by [65] have relatively homogeneous water content (~0.2 wt. %) and δD values (~1000‰), slightly shifting to a D-poor end-member. Chen et al. [60] reported the impact melt glasses in Tissint having 179–2388 ppm water and δD values of 45–4867‰, displaying a positive correlation (Figure 5). Two spots from EETA 79001A have ~0.1 wt. % water and δD values of ~1500‰ [72], close to the dataset reported by Hallis et al. [65] in Tissint. In contrast, the dataset reported by Liu et al. [71] in EETA 79001C share a similar trend to that reported by Chen et al. [60]. ALH 77005, a poikilitic shergottite, has 105–520 ppm water and δD values of 1301–3030‰ [59]. LAR 06319 has relatively homogenous water content (118–164 ppm) and δD values (~2100–2900‰) in impact melt glasses [72]. Except for three spots from Tissint [65] and two spots from EETA 79001A [72], the other analytical results of

impact melt glasses also plot along the two end-member mixing trend similar to the melt inclusion glasses and maskelynites (Figure 1, Figure 4, and Figure 5) [51,58–61,63,65,67,69–73]. This suggests that water was retained in impact melt glasses from the Martian crustal/surficial water reservoir. Such a scenario is further supported by the zonation features of $H_2O$ and $CO_2$ cross-cutting shock melt veins in Tissint [60], as well as a positive relationship between $H_2O$ with both Cl and S along the mixing of Martian mantle and surficial materials [60]. The negative correlation between water content and δD values in shock melt phases in Tissint could be explained by either degassing during shock or by increasing amounts of epoxy contamination [65].

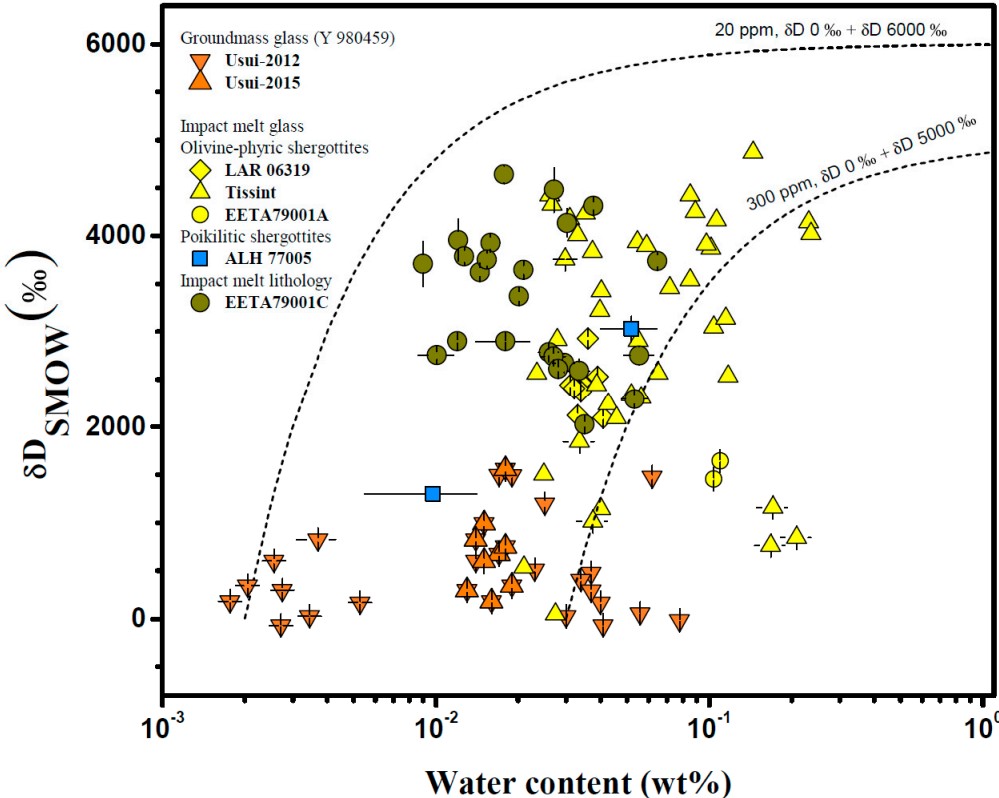

**Figure 5.** The relationship between water content and δD values of groundmass glasses and impact melt glasses from Martian shergottites. Most datasets are plotting along a two end-member mixing trend except for some spots shifting towards to higher water content. Groundmass glasses of Y 980459 from [53,72]. Impact melt glasses: LAR 06319 (LAR06) and EETA 79001A (EETA-A) from [72], Tissint from [60,65,70], EETA 79001C from [71], ALH 77005 (ALH) from [59].

　　Groundmass glasses from olivine-phyric shergottites, such as Y-980459, probably quenched from the parent melt [127]. They have 18–800 ppm water content and δD values of −71–1562‰ (assuming $H_2O$ = H/Si without absolute concentration calibration) [53,72]. The water content (100–200 ppm) of the groundmass glasses in Y 980459 positively correlate with δD values (181–1562‰) [72]. Usui et al. [72] proposed an intermediate D-enriched Martian surficial water reservoir with a δD value of ~3000‰, which interacted with groundmass glasses in Y 980459 on Mars. In comparison, the earlier results of groundmass glasses reported by Usui et al. [53] have larger variations in water content (~20–800 ppm) than their later studies. These results demonstrate the complexity in the relationship between $H_2O$ and δD, a weak, positive trend when $H_2O$ <200 ppm and a negative trend when $H_2O$ > 200 ppm. The high-$H_2O$ spots were not duplicated in their later works [72] perhaps relating to improvements made in avoiding terrestrial contamination as supported by the significant lower C/Si ratios [53,72]. Indeed, most analytical spots of groundmass glasses from Y 980459 plot along a two end-member mixing trend similar to impact melt glasses (Figure 5). This is likely the result of water-rock interactions on the

near surface of Mars. Several high-H₂O and low-δD spots could be explained by either the result of $H_2O$ heterogeneity of the analytical target or a higher contribution from terrestrial contamination (e.g., covering some micro-cracks during analysis).

*4.6. Nominal Anhydrous Minerals/Phases*

Olivine and Pyroxene: The water content of olivine and pyroxene in Tissint, Shergotty, Zagami, EETA 79001A, EETA 79001C, and ALH 77005 [51,52,59,65,71] vary from ~1 to 1400 ppm and negatively correlate with the δD values (Figure 6), representing a D-poor (δD ~ 0‰) end-member interacting with a H₂O-poor and D-rich (δD up to 3000‰) end-member either on Mars or Earth. This is either probably the result of water degassing during crystallization [76,97], varying degrees of epoxy contamination [65], or terrestrial alteration [99]. Degassing of water on Mars cannot account for the large fractionation in hydrogen isotopic compositions up to ~3000‰ under the current Martian conditions [76]. It requires a secondary geologic event to explain the δD values of olivine and pyroxene to ~3000‰ because they should equilibrate with the parent magma in hydrogen isotopic compositions (~0‰) during crystallization [51–53]. Liu et al. [71] reported that olivine enclosed in an impact melt from EETA 79001C with water content from 4 to 7 ppm and δD values from −146 to −55‰, suggesting that heavy shock metamorphism on Mars did not alter the H₂O and δD values of olivine although the impact melt glasses were enriched in D. In contrast, pyroxene in or in close contact to an impact melt from EETA 79001C has 10 to 41 ppm water with δD values varying from 1729 to 2837‰, significantly higher than that of olivine [71]. These features indicate that the shock metamorphism recorded in those Martian meteorites redistributed H in the mafic minerals, at least for pyroxene and impact melt glasses [71]. There also requires additional geologic event(s) in order to increase the hydrogen isotopic composition of the bulk samples prior to shock metamorphism, which is consistent with the significant D-enrichment signatures of the shergottites by step-heating measurements [89].

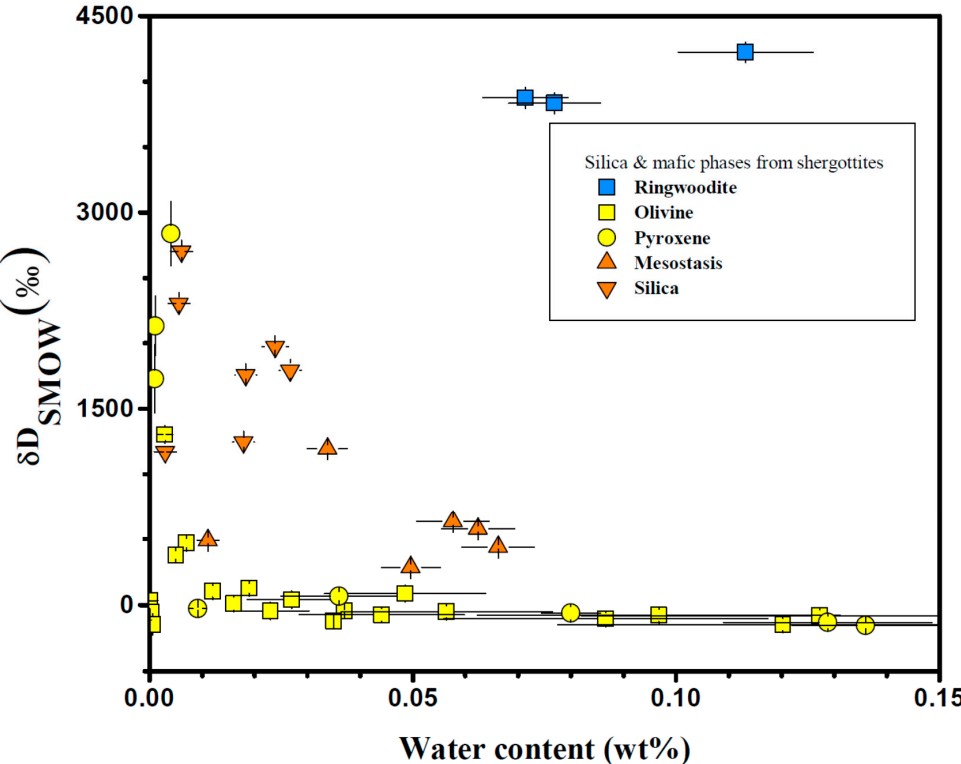

**Figure 6.** The relationship between water content and δD values of silica and mafic phases from Martian shergottites. Ringwoodite found in Tissint from [65], post-stishovite (silica) and mesostasis of Shergotty and Zagami from [59], Olivine and pyroxene of Tissint, Shergotty, Zagami, EETA 79001A & C, and ALH 77005 from [51,52,59,65,71].

Post-stishovite, Mesostasis and Ringwoodite: The water content and δD values of post-stishovite (shock induced origin [126]) and mesostasis from Shergotty and Zagami vary from 56 to 662 ppm and from 285 to 2704‰ [59] respectively, showing a linear negative correlation except for two spots located along the trend of olivine and pyroxene (Figure 6). Post-stishovite has lower water content and higher δD values than that of mesostasis (Figure 6), suggesting that the mesostasis either retained more of the Martian mantle signatures or was terrestrially contaminated, whereas D-rich features of post-stishovite attributes to shock metamorphism on Mars [59]. Ringwoodite (a high-pressure polymorph of olivine) from Tissint shows high δD values and a weak positive correlation between δD values and water content, suggesting D-rich Martian crustal water retainment during the shock event [65]. The significant difference between water content in post-stishovite and ringwoodite is probably dependent on their water capacity as well as pressure, temperature, and cooling rates during shock metamorphism [65].

## 5. Summary and Outlook

The D-rich signatures of melt inclusions, apatite, merrillite, maskelynite, IMG & GG, and nominal anhydrous phases from various shergottites suggest a contribution of D in water from Martian crustal/surficial water that H isotopically equilibrated with the Martian atmosphere [55–57]. The diffusional profiles of melt inclusions, maskelynite, and IMG support the diffusion of D in water post-crystallization, rather than from the crystallization of the magma. However, the abundance of water and duration of the water-rock interaction recorded in the shergottites could not have been sufficient enough to result in mineral alteration [67]. Deuterium-rich water laths in melt inclusions [68] are examples of the limits of occurrences of most water in the shergottites [83–88] representative of the cold and dry paleoclimate encountered by the shergottite lithology on Mars [6,38,48].

$H_2O$ and δD in melt inclusions, apatite, maskelynite, and nominal anhydrous phases plot along two distinct populations. One is a positive trend, consistent with post-crystallization interaction of the Martian surficial D-enriched water (δD ~ 5000–6000‰ with a Martian mantle signature (δD ~ 0‰) assuming no significant D fractionation during crystallization stages) [53,60,67,68,70–72] (Figure 1, Figure 2, Figure 4, and Figure 5). The other one is a negative trend, which requires alternative geologic event(s) increasing δD from ~0‰ of Martian magmatic signature to thousands of per mil of Martian surficial water signature prior to an addition of D-poor water to increase $H_2O$ and decrease δD either by interaction of degassing water on Mars [51] or terrestrial contamination [52,99]. It is still a challenge to constrain the place where the D-poor water came from. However, recent progresses on C [49], O [78], Cl [79–81,102], and S [82] isotopic compositions, as well as the elemental abundance of C, F, S, and Cl [53,60,68,70–72] could shed more light on the nature of the volatile inventories on Mars.

**Author Contributions:** Writing—original draft preparation, S.W. and S.H.; visualization, S.W. and S.H.; supervision, S.H.; funding acquisition, S.H. All authors have read and agreed to the published version of the manuscript.

**Funding:** This research was funded by the National Natural Science Foundation (grant numbers 41573057, 41430105, and 41490631) and the Key Research Program of the Institute of Geology & Geophysics, CAS (grant number IGGCAS-201905).

**Acknowledgments:** We thank three anonymous reviewers for their constructive comments and suggestions, and A. E. Hitesh Changela and Elias Chatzitheodoridis for invitation of this review, handling this manuscript, and language polishing.

**Conflicts of Interest:** The authors declare no conflict of interest.

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
