# Peer review of "Hydrogen Isotopic Variations in the Shergottites"

_geosciences, doi:10.3390/geosciences10040148_

Round 1

Reviewer 1 Report

Wang and Hu presents a review of hydrogen isotopic measurements in shergottites. The revision is quite improved from the original manuscript, but there are still major issues with the review of the petrology of shergottites that are factually incorrect and need to be corrected before publication. Further, the manuscript relies on an outdated review paper (Bridges and Warren/Bridges et al.) for much of the literature review. That review is fine but is now over a decade old and much has changed. Therefore, I beg the authors to do a full review on their own and not rely heavily on an outdated, and sometimes incorrect, review paper. I have included many missing references below, but I truly implore the authors to do their own literature search, as that is not the job of a reviewer. Finally, I have put the sections in bold where the manuscript is factually incorrect, and it is vital that the authors revise these accordingly.

Line 34: the term lherzolitic was used original but has now been changed to poikilitic (e.g., Walton et al. 2012; Combs et al. 2019 but please do a reference search for other missing important references). The reason for this is that they are cumulate igneous rocks and not peridoitic as the manuscript states. Therefore, please update the nomenclature to include the new term and remove peridotite from the manuscript as it is incorrect.

Lines 35-37: please combined these two sentences. Otherwise, it looks like these are separate thoughts when in fact, the ‘highly depleted’ are called depleted in the vast majority of the Martian meteorite literature. Further, please use other missing references here such as: (Laul et al. 1986; Longhi 1991; Norman 1999; Borg and Draper 2003; Jones 2003; Filiberto et al. 2012; Jones 2015; Howarth and Udry 2017).

Lines 40-42: This correlation now does not exist with the newer data we have, especially once the poikilitic shergottites are included; therefore, this is an outdated model and the manuscript should at least acknowledge this (e.g., Howarth et al. 2014 see extensive work by Arya Udry and coauthors on this topic; Howarth and Udry 2017).

Lines 43-44: some clasts in the rock are 4.4 Ga, others are 2.1Ga. This is an incredibly complex rock and cannot be summarized as simply as is done here (e.g., McCubbin et al. 2016).

Line 46: Most shergottites are 180 Ma (Jones 2007), but we now have two that are ~2 billion years old (Agee et al. 2014; Herd et al. 2017; Lapen et al. 2017). These were originally not mentioned in this review.

Lines 108-112: The debate about the water content of shergottites is largely over. The community has coalesced around dry Martian magmatism (e.g., Filiberto et al. 2016 and other articles in this special issue).

Lines 118-119: While Bridges et al. did review the secondary minerals, this is incredibly outdated. Please update and revise this review.

Lines 150-151: This sentence is factually incorrect. I pointed this out previously but will point it out again: Melt inclusions do not retain the host magmatic composition. This must be changed in the manuscript! They react with the wall olivine to change the major element bulk composition and water can easily diffuse in and out of the host. Please change this accordingly. Here are many references showing this: (Sobolev 1996; Danyushevsky et al. 2000; Gaetani and Watson 2000; Danyushevsky et al. 2002; Baker 2008; Calvin and Rutherford 2008; Portnyagin et al. 2008; Ruscitto et al. 2011; Bucholz et al. 2013; Goodrich et al. 2013). I could find easily twice this number of references showing the issues of dealing with melt inclusions as they do not fully retain the most magmatic composition at all, but can be used to deduce this, with some assumptions. Instead, the manuscript references one abstract and not a peer-reviewed manuscript.

Line 183 (and throughout) change lherzolitic to poikilitic as these are not peridotites.

Line 263: Merrillite cannot be a water-free mineral if you can measure the water content. By definition this contradicts with the manuscript.

Line 393-403: These are shock minerals and not igneous minerals (e.g., Walton and Spray 2003; Walton et al. 2012; Walton et al. 2014). They are an interesting comparison, but this section needs to address this.

Section 5: I recommended cutting this section previously and the revision has only convinced me more that this section needs to be cut – especially both figures. There is no petrologic basis for this comparison. Each of these minerals and materials takes water at different rates, which is why we have partition coefficients; therefore, comparing the bulk based on counts between these different materials is scientifically invalid. Unfortunately, the authors ignored this comment in the revision but this section is petrologically flawed and cannot be published without a petrologic reasoning for such a comparison.

Agee, C.B., Muttik, N., Ziegler, K., McCubbin, F., Herd, C., Rochette, P., Gattacceca, J., 2014. Discovery of a New Martian Meteorite Type: Augite Basalt---Northwest Africa 8159. Lunar and Planetary Science Conference 45, Abstract# 2036.

Baker, D., 2008. The fidelity of melt inclusions as records of melt composition. Contributions to Mineralogy and Petrology 156, 377-395.

Borg, L.E., Draper, D.S., 2003. A petrogenetic model for the origin and compositional variation of the martian basaltic meteorites. Meteoritics & Planetary Science 38, 1713-1731.

Bucholz, C.E., Gaetani, G.A., Behn, M.D., Shimizu, N., 2013. Post-entrapment modification of volatiles and oxygen fugacity in olivine-hosted melt inclusions. Earth and Planetary Science Letters 374, 145-155.

Calvin, C., Rutherford, M.J., 2008. The parental melt of lherzolitic shergottite ALH 77005: A study of rehomogenized melt inclusions. American Mineralogist 93, 1886-1898.

Combs, L.M., Udry, A., Howarth, G.H., Righter, M., Lapen, T.J., Gross, J., Ross, D.K., Rahib, R.R., Day, J.M., 2019. Petrology of the enriched poikilitic shergottite Northwest Africa 10169: Insight into the martian interior. Geochimica et Cosmochimica Acta 266, 435-462.

Danyushevsky, L.V., Della-Pasqua, F.N., Sokolov, S., 2000. Re-equilibration of melt inclusions trapped by magnesian olivine phenocrysts from subduction-related magmas: petrological implications. Contributions to Mineralogy and Petrology 138, 68-83.

Danyushevsky, L.V., McNeill, A.W., Sobolev, A.V., 2002. Experimental and petrological studies of melt inclusions in phenocrysts from mantle-derived magmas: an overview of techniques, advantages and complications. Chemical Geology 183, 5-24.

Filiberto, J., Baratoux, D., Beaty, D., Breuer, D., Farcy, B.J., Grott, M., Jones, J.H., Kiefer, W.S., Mane, P., McCubbin, F.M., Schwenzer, S.P., 2016. A review of volatiles in the Martian interior. Meteoritics & Planetary Science 51, 1935-1958.

Filiberto, J., Chin, E., Day, J.M.D., Franchi, I.A., Greenwood, R.C., Gross, J., Penniston-Dorland, S.C., Schwenzer, S.P., Treiman, A.H., 2012. Geochemistry of intermediate olivine-phyric shergottite Northwest Africa 6234, with similarities to basaltic shergottite Northwest Africa 480 and olivine-phyric shergottite Northwest Africa 2990. Meteoritics & Planetary Science 47, 1256-1273.

Gaetani, G.A., Watson, E.B., 2000. Open system behavior of olivine-hosted melt inclusions. Earth and Planetary Science Letters 183, 27-41.

Goodrich, C.A., Treiman, A.H., Filiberto, J., Gross, J., Jercinovic, M., 2013. K2O-rich trapped melt in olivine in the Nakhla meteorite: Implications for petrogenesis of nakhlites and evolution of the Martian mantle. Meteoritics & Planetary Science 48, 2371-2405.

Herd, C.D.K., Walton, E.L., Agee, C.B., Muttik, N., Ziegler, K., Shearer, C.K., Bell, A.S., Santos, A.R., Burger, P.V., Simon, J.I., Tappa, M.J., McCubbin, F.M., Gattacceca, J., Lagroix, F., Sanborn, M.E., Yin, Q.-Z., Cassata, W.S., Borg, L.E., Lindvall, R.E., Kruijer, T.S., Brennecka, G.A., Kleine, T., Nishiizumi, K., Caffee, M.W., 2017. The Northwest Africa 8159 martian meteorite: Expanding the martian sample suite to the early Amazonian. Geochimica et Cosmochimica Acta 218, 1-26.

Howarth, G.H., Pernet-Fisher, J.F., Balta, J.B., Barry, P.H., Bodnar, R.J., Taylor, L.A., 2014. Two-stage polybaric formation of the new enriched, pyroxene-oikocrystic, lherzolitic shergottite, NWA 7397. Meteoritics and Planetary Science 49, 1812-1830.

Howarth, G.H., Udry, A., 2017. Trace elements in olivine and the petrogenesis of the intermediate, olivine-phyric shergottite NWA 10170. Meteoritics & Planetary Science 52, 391-409.

Jones, J., 2003. Constraints on the structure of the martian interior determined from the chemical and isotopic systematics of SNC meteorites. Meteoritics & Planetary Science 38, 1807-1814.

Jones, J.H., 2007. The Shergottites are Young. Meteoritics & Planetary Science 42, 5194.

Jones, J.H., 2015. Various aspects of the petrogenesis of the Martian shergottite meteorites. Meteoritics & Planetary Science 50, 674-690.

Lapen, T.J., Righter, M., Andreasen, R., Irving, A.J., Satkoski, A.M., Beard, B.L., Nishiizumi, K., Jull, A.J.T., Caffee, M.W., 2017. Two billion years of magmatism recorded from a single Mars meteorite ejection site. Science Advances 3, DOI: 10.1126/sciadv.1600922.

Laul, J.C., Smith, M.R., Wänke, H., Jagoutz, E., Dreibus, G., Palme, H., Spettel, B., Burghele, A., Lipschutz, M.E., Verkouteren, R.M., 1986. Chemical Systematics of the Shergotty Meteorite and the Composition of Its Parent Body (Mars). Geochimica et Cosmochimica Acta 50, 909-926.

Longhi, J., 1991. Complex magmatic processes on Mars-Inferences from the SNC meteorites. Proceedings of the Lunar and Planetary Science Conference 21, 695-709.

McCubbin, F.M., Boyce, J.W., Novák-Szabó, T., Santos, A.R., Tartèse, R., Muttik, N., Domokos, G., Vazquez, J., Keller, L.P., Moser, D.E., Jerolmack, D.J., Shearer, C.K., Steele, A., Elardo, S.M., Rahman, Z., Anand, M., Delhaye, T., Agee, C.B., 2016. Geologic history of Martian regolith breccia Northwest Africa 7034: Evidence for hydrothermal activity and lithologic diversity in the Martian crust. Journal of Geophysical Research: Planets 121, 2120-2149.

Norman, M., 1999. The composition and thickness of the crust of Mars estimated from REE and Nd isotopic compositions of Martian meteorites. Meteoritics & Planetary Science 34, 439-449.

Portnyagin, M., Almeev, R., Matveev, S., Holtz, F., 2008. Experimental evidence for rapid water exchange between melt inclusions in olivine and host magma. Earth and Planetary Science Letters 272, 541-552.

Ruscitto, D., Wallace, P., Kent, A., 2011. Revisiting the compositions and volatile contents of olivine-hosted melt inclusions from the Mount Shasta region: implications for the formation of high-Mg andesites. Contributions to Mineralogy and Petrology 162, 109-132.

Sobolev, A., 1996. Melt inclusions in minerals as a source of principle petrological information. Petrology 4, 209-220.

Walton, E., Irving, A., Bunch, T., Herd, C., 2012. Northwest Africa 4797: A strongly shocked ultramafic poikilitic shergottite related to compositionally intermediate Martian meteorites. Meteoritics & Planetary Science 47, 1449-1474.

Walton, E.L., Sharp, T.G., Hu, J., Filiberto, J., 2014. Heterogeneous mineral assemblages in martian meteorite Tissint as a result of a recent small impact event on Mars. Geochimica et Cosmochimica Acta 140, 334-348.

Walton, E.L., Spray, J.G., 2003. Mineralogy, microtexture, and composition of shock-induced melt pockets in the Los Angeles basaltic shergottite. Meteoritics & Planetary Science 38, 1865-1875.

Reviewer 2 Report

The manuscript has been well improved.

I have minor comments to address.

Line 58 – Hesperian: add the age in brackets

Line 60-61 “This can be explained by the loss of Martian water to space via solar wind bombardment during the decay of its magnetic field 59 [23], enriching its atmosphere in deuterium (D = 4950 ± _1080 ‰ [24]) relative to the initial D of ~ 60 0 ‰ [25-28]H [29]. This observation is consistent with amongst the highest measured D/H ratios (D 61 up to ~7000 ‰) for the Martian atmosphere by FTIR spectrometry [29,30].”

 (25-28) H (29): some typos here.

This sentence is not very clear, as you said that the atmosphere is enriched, and then, that is consistent with High D values.

You could write, “…enriching its atmosphere in deuterium”, as observed by in situ analyses (D = 4950 ±_1080 ‰ [24]) as well as FTIR spectrometry of the Martian atmosphere (29,30).

Line 65 – “A meteorite parent body should equilibrate with its local hydrogen isotopic composition during crystallization (~0 ‰, [25-28]).”

I would replace by “A meteorite should equilibrate with its parent body hydrogen isotopic composition during crystallization (~0 ‰, [25-28]).”

Line 73 -  “tend to support that isotopic exchanges occurred on the near surface of Mars” Between Martian minerals and  ?

Line 156 –“ would also affect the water content and the H isotopic composition in the melt inclusions in non-mutually exclusive ways”

Line 160 – “ALH 77005 at 1 ± 0.3 to 1770 ± 200 ppm” is it 0.1 like in the table or the other way around?

Line 213 – change Moreover by However

Line 238 – “probably overlapped adjacent small grains of mafic minerals such as apatite” Are you sure you mean apatite and not pyroxene or olivine? Doesn’t make sense that a apatite spot overlaps another apatite.

Line 282 – “Alternatively, 280 such distributions can be explained by a mixing of Martian mantle signature (D ~ 0 ‰) with D-281 enriched Martian crustal water reservoir when ignoring those spots with significant high water 282 contents and low D values (Figure 3).”

So in this case you should get a positive trend (if you remove the sports with high H20), similar to what is observed for Mi and for apatites right?

Because if the trend is negative, it is difficult to advocate for a similar processus that form positive trends for the other minerals. The negative trend is then well explained by the dehydration processus.

Line 313 – correct 4-40 by 20-80 ppm

Also in figure 4, you should have the same legend format as the other graphs.

Line 336 – “impact melt glasses also plot along the two end-member 335 mixing trend similar to the melt inclusions (Figure 5)” also similar to maskelynites and some extent apatites.

Line 351 – rephrase “it measures”

Chapter 5 – One thing that you could discuss here is the result from the modelling curves of mixing you plot for MI, apatites, maskelynites, IMC and groundmass glasses. From these models, you start the initial reservoir with a composition of 0 permil, and between 4 to 300 ppm, function of the minerals. Is these values give you insights into the water content of the parent melt? How does it relate to previous water estimation in shergottite parent melt? For instance, Mi are direct measurements of the parent melt signature.

Round 2

Reviewer 1 Report

The revised manuscript is vastly improved from the previous two iterations. The authors have done a fantastic job revising the manuscript to make it a coherent review of the literature. However, since this was last reviewed, a new manuscript on D/H of Martian materials has been published. This manuscript focuses more on Noachian Mars based on the breccia samples (Barnes et al. 2020), but does include a discussion of these reservoirs through time. Therefore, a discussion of this work needs to be included here for completeness. Once that is added and discussed, I see no impediment to publishing this work.

Link to new paper:

https://www.nature.com/articles/s41561-020-0552-y?proof=true&draft=collection

Author Response

This manuscript is a resubmission of an earlier submission. The following is a list of the peer review reports and author responses from that submission.

Round 1

Reviewer 1 Report

This paper reviews the recent progresses in hydrogen isotope analyses (especially, in-situ analyses by SIMS) of various water-bearing/nominally anhydrous phases in shergottites, the youngest group of Martian meteorites. The manuscript seems to be well organized and cover the most important studies. I think this work is worth to be published after the following minor modifications. 

- You should also refer a recent review by Dr. T. Usui (2019), Hydrogen reservoirs in Mars as revealed by Martian meteorites, in "Volatiles In The Martian Crust" (Eds. Filiberoto J. and Schwenzer S. P.), which may have close relationship to this paper.  

- The word "SNC" in the title might be misleading, because the other studies about "NCs" are not reviewed here in detail. I suggest you to either modify the title or add a (sub)section that summarizes the D/H - H2O relations of the other Martian meteorites. 

Because I am not a native English speaker, I cannot say anything about language/grammar. Please follow other native comments and/or commercial English proof. 

I added other specific comments/questions to the manuscript by Adobe Reader's Note tool (attached PDF). Please check it. 

Reviewer 2 Report

Wang and Hu present a review of the water content of minerals and glasses within shergottite meteorites. A review of this kind hasn’t been done recently and could be an important manuscript for the community. Unfortunately, this is not that manuscript. The manuscript is poorly written, does not know the literature, and often mixes igneous and secondary signatures (equating all signatures with secondary processing). Further, the manuscript is incredibly short for a review paper and leaves the reader wondering why publish this. It reads as a data dump of review data with no new conclusions, open questions, or future directions. Therefore, I cannot recommend publication.

Some thoughts that are missing:

Table 1 is not a comprehensive list of hydrous phases in shergottites. Almost all have amphibole, even if no H- or D/H measurements have been made, which makes this table relatively useless.

There are many missing papers and ideas. For examples, McCubbin and Barnes (2019; EPSL) suggests that the D/H variability has nothing to do with alteration, which could make this paper incorrect. This idea is not even discussed.

Water and D/H diffusion in melt inclusions is actually quicker than in minerals where water is structurally bound (see extensive work by Glenn Gaetani and many others in the terrestrial literature), but this is not discussed. Further, if a magmatic signature is detected in melt inclusions, that would negate any secondary processes – as proposed here.

The manuscript acknowledges that D/H is not always correlated with other geochemical signatures (level of REE enrichment, sulfur or chlorine isotopes, or total volatile content), but does not explore what this means or why. For a useful review paper, this is vital.

The manuscript incorrectly discusses the REE enrichment in bulk shergottites. It should be depleted, intermediate, and enriched. Not very depleted, depleted, and slightly depleted (in fact, the REE patterns are relatively flat or have a slight uptick and so are in fact not depleted at all).

Currently all secondary minerals in shergottites are thought to be terrestrial alteration. We do not have definitive proof that any are from Mars. We do have definitive proof in nakhlites, chassignites, and alh84001, but not shergottites. This needs to be heavily revised.

Figures: The symbols and colors are not consistent throughout. Sometimes meteorites are lumped together, others each meteorite has its own symbol. For a review paper, this needs to be streamlined and consistent for ease of reading.

Figure 7 is relatively useless. It is impossible to see any trends. Further, there should be no comparison of bulk water or D/H ratios between minerals and glasses, because of partitioning (during crystallization, impact, or other secondary processes) each will have a different value and cannot be compared on a simple correlation diagram.

Reviewer 3 Report

This paper aims to give a review of water-rock interactions on Mars in the shergottites. While, the chapters detailing each martian phase are a bit better, this paper lacks a lot of references (in the introduction and chapter 2), data, and discussion. Moreover, a lot of sentences seem unclear and too long. This paper appears to me as a list of facts with no substantial scientific argumentation to support these facts.

For instance, while this paper talks about hydrogen isotopic composition (mentioned in the first line of the abstract), the first actual delta D value is only given line 102. The introduction and chapter 2 discussed about D-enrichments without stating what these enrichments are relative to or what the actual values we are talking about are. For instance, the paper states that Martian meteorites are D-enriched but no ranges of delta D is given (as well as no references). The Table 1 “A summary of water contents and H isotope analysis of shergottites” gives no values of H2O or dD, which should be contained in this table. 

The paper also referred a lot to D-poor or D-rich end-members with no implication of what could be these end-members. In a same way,the paper stated that some martian phases have been D-enriched by mixing with a D-rich reservoir but no discussion at all is given on what could represent the initial/unaltered dD value of the mineral and what was this initial dD signature (of the Martian mantle).

In fact some portions of the text are unclear, no justification or argumentation are given. The whole paper lacks discussion of the facts enunciated. For example, the section 4.1 has only few discussion on how the results presented support the water-rock interactions in Shergottites and looks more like a list of H2O and dD measurements.  Moreover, some portions of text are contradictory. For example line 159 “Tissint apparently plot along a reverse trend between H2O and dD values, contrary to that of GRV 020090 (Fig.1)” while line 162 “However, [61] reported the melt inclusions from Tissint having similar trends to that from GRV 020090”. But these “new” data are not reported on the Fig. 1. Again, no discussion on these contradictory results.

The lack of reference is also a big problem in the introduction and chapter 2. As an example, lines 38-40, this paper stated that the Martian meteorites recorded D-enrichments through a variety of geological processes. No references are given to support the fact that the Martian meteorites are D-enriched or to support the various geological processes referred to. This is too weak for a review paper. Lot of references are appearing later in the text but they should also appear in the introduction and chapter 2.

As such, I suggest that a lot of rewriting needs to be done before this paper can be accepted for publication. I made some comments on the pdf version, but more rewriting, a lot more discussion, have to be done than just these comments.
